# PAC Privacy Preserving Diffusion Models

## Abstract

Data privacy protection is garnering increased attention among researchers. Diffusion models (DMs), particularly with strict differential privacy, can potentially produce images with both high privacy and visual quality. However, challenges arise such as in ensuring robust protection in privatizing specific data attributes, areas where current models often fall short. To address these challenges, we introduce the PAC Privacy Preserving Diffusion Model, a model leverages diffusion principles and ensure Probably Approximately Correct (PAC) privacy. We enhance privacy protection by integrating a private classifier guidance into the Langevin Sampling Process. Additionally, recognizing the gap in measuring the privacy of models, we have developed a novel metric to gauge privacy levels. Our model, assessed with this new metric and supported by Gaussian matrix computations for the PAC bound, has shown superior performance in privacy protection over existing leading private generative models according to benchmark tests.

## 1 Introduction

Modern deep learning models, fortified with differential privacy as defined by Dwork et al. (2006), have been instrumental in significantly preserving the privacy of sensitive data (Dwork et al., 2014). DP-SGD Abadi et al. (2016), a pioneering method for training deep neural networks within the differential privacy framework, applies gradient clipping at each step of the SGD (stochastic gradient descent) to enhance privacy protection effectively.

Diffusion models (DMs) (Song & Ermon, 2019; 2020; Dhariwal & Nichol, 2021) have emerged as state-of-the-art generative models, setting new benchmarks in various applications, particularly in generating high-quality images. When trained under strict differential privacy protocols, these DMs can produce images that safeguard privacy while maintaining high visual fidelity. For instance, DPGEN (Chen et al., 2022) leverages a randomized response technique to privatize the recovery direction in the Langevin MCMC process for image generation. The images produced by DPGEN are not only visually appealing but also compliant with differential privacy standards, although its privacy mechanism has been shown to be data-dependent later on. Moreover, the Differentially Private Diffusion Models (DPDM) (Dockhorn et al., 2022) adapt DP-SGD and introduce noise multiplicity in the training process of diffusion models, demonstrating that DPDM can indeed produce high-utility images while strictly adhering to the principles of differential privacy.

While diffusion models integrated with differential privacy (DP) mark a significant advance in privacy-preserving generative modeling, several challenges and limitations remain.

- Most research on diffusion models with DP has concentrated on the privatization of overall image features. The need to privatize specific attributes, such as facial expressions in human portraits, has not been adequately addressed. This oversight suggests a gap in the nuanced application of DP in generative modeling.

- The core objective of DP is to evaluate the distinguishability between processed and original data. Yet, a significant hurdle is the difficulty of providing adversarial worst-case proofs, especially when comparing two distinct datasets. This difficulty presents a substantial barrier to validating the efficacy of DP methods in practical scenarios.

- The absence of a robust privacy measurement for models poses a critical challenge. Without a clear metric, it becomes problematic to assess and compare the data privacy protection performance across different models. This lack of standardized evaluation complicates the advancement and adoption of privacy-preserving techniques in the field.

These issues highlight the need for continued research and development to overcome the current limitations of diffusion models with DP and to push the boundaries of privacy protection in generative modeling.

Recently, Xiao & Devadas (2023) introduces a novel definition of privacy known as Probably Approximately Correct (PAC) Privacy, representing a significant evolution in privacy-preserving methodologies. This definition diverges from traditional Differential Privacy (DP) by focusing on the difficulty of reconstructing data using any measure function, thereby offering broader applicability beyond the adversarial worst-case scenarios considered by DP. Regarding utility, the necessary perturbation magnitude in PAC Privacy is not necessarily constrained by DP lower bound of $\Theta(\sqrt{d})$ for a $d$-dimensional release. Instead, it could be as low as $O(1)$ for many practical data processing tasks. This stands in contrast to the input-independent worst-case information-theoretic lower bound. Furthermore, leveraging PAC privacy comes with a framework that autonomously determines the minimal noise addition necessary for effective data protection (Xiao & Devadas, 2023). This framework simplifies the process of achieving privacy-preserving data handling, making it more accessible for widespread use in the field of data science and beyond.

To tackle the aforementioned challenges, we have introduced PAC Privacy Preserving Diffusion Models (P3DM). Drawing from the foundations of DPGEN and harnessing insights from conditional classifier guidance (Dhariwal & Nichol, 2021; Batzolis et al., 2021; Ho & Salimans, 2022), our P3DM incorporates a conditional private attribute guidance module during the Langevin sampling process. This addition empowers the model to specifically target and privatize certain image attributes with greater precision.

Furthermore, we have crafted a set of privacy evaluation metrics. These metrics operate by measuring the output class labels of the two nearest neighbor images in the feature space of the Inception V3 model (Szegedy et al., 2016), using a pretrained classifier. Additionally, we quantify the noise addition $B$ necessary to assure PAC privacy in our model and conduct comparative analyses against the mean L2-norm of $B$ from various other models.

Through meticulous evaluations that utilize our privacy metrics and benchmarks for noise addition, our model has proven to offer a superior degree of privacy. It exceeds the capabilities of state-of-the-art (SOTA) models in this critical aspect, while simultaneously preserving the high quality of synthetic image samples. These samples remain on par with those produced by the state-of-the-art models, illustrating that our model does not sacrifice quality for privacy. This achievement underscores our model's potential to set new precedents in the domain of privacy-preserving image generation and data protection at large.

Our contributions are summarized as follows:

- We propose the first diffusion model with analysis on its PAC privacy guarantees.

- We incorporate conditional private classifier guidance into the Langevin Sampling Process, enhancing the protection of privacy for specific attributes in images.

- We introduce a new metric that we developed for assessing the extent of privacy provided by models.

- We compute the noise addition matrix to establish the Probably Approximately Correct (PAC) upper bound and have conducted a comparative analysis of the norm of this matrix across various models.

- Through extensive evaluations, we demonstrate that our model sets a new standard in privacy protection of specific attributes, achieving state-of-the-art (SOTA) results, while maintaining image quality at a level that is comparable to other SOTA models.

## 2 Related Work

### 2.1 Early Works on Differentially Private Image Generation

Image Synthesis with differential privacy has been studied extensively during the research. Recently, there has been an increased emphasis on utilizing sophisticated generative models to improve the quality of differentially private synthetic data (Hu et al., 2023). Some approaches employ Generative Adversarial Networks (GANs) (Goodfellow et al., 2014), or GANs that have been trained using the Private Aggregation of Teacher Ensembles (PATE) framework (Xie et al., 2018; Chen et al., 2020; Harder et al., 2021; Torkzadehmahani et al., 2019). Other contributions leverage variational autoencoders(VAEs) (Pfitzner & Arnrich, 2022; Jiang et al., 2022; Takagi et al., 2021), or take advantage of customized architectures (Cao et al., 2021; Harder et al., 2023). However, there are several limitations for those DP synthesizers: (1) Failure when applying to high-dimensional data, primarily due to the constraints imposed by discretization. (2) Limited image quality and lack of expressive generator networks (Cao et al., 2021).

### 2.2 Differentially Private Diffusion Models

Diffusion models (DMs) (Song & Ermon, 2019; 2020; Dhariwal & Nichol, 2021) , recognized for setting new standards in image generation, can produce high-quality, privacy-compliant images when trained with differential privacy protocols. For example, DPGEN (Chen et al., 2022) employs a data-dependant randomized response method to privatize the recovery direction in the Langevin MCMC process for image generation. Furthermore, Differentially Private Diffusion Models (DPDM) (Dockhorn et al., 2022), which adapt DP-SGD and introduce noise multiplicity, both demonstrate the feasibility of generating visually appealing, privacy-protective images. Subsequent advancements, including fine-tuning existing models and employing novel diffusion model architectures, have been made to boost the effectiveness of differentially private image generation, as detailed in Ghalebikesabi et al. (2023); Lyu et al. (2023). Nevertheless, as previously noted in the introduction, there remains three key challenges to be addressed. In the following sections, we propose solutions and methods to tackle these issues.

## 3 Background

### 3.1 Differential Privacy

A randomized mechanism $\mathcal{M}$ is said $(\varepsilon, \delta)$- differentially private if for any two adjacent datasets $D$ and $D'$ differing in a single datapoint for any subset of outputs $S$ as follows (Dwork et al., 2014):

$$\mathbf{Pr}[\mathcal{M}(D) \in S] \leq e^{\varepsilon} \cdot \mathbf{Pr}[\mathcal{M}(D') \in S] + \delta \tag{1}$$

Here, $\varepsilon$ is the upper bound on the privacy loss corresponding to $\mathcal{M}$, and $\delta$ is the probability of violating the DP constraint.

Differential privacy is a mathematical approach designed to protect individual privacy within datasets. It offers a robust privacy assurance by enabling data analysis without disclosing sensitive details about any specific person in the dataset.

### 3.2 PAC Privacy

Traditional provable privacy methods like Differential Privacy (DP) (Dwork et al., 2014), often require strong assumptions, which can be precisely computed only in a few simple applications, such as aggregation or linear queries, and can lead to significant accuracy loss on utility. What's more, the absence of robust risk quantification tools significantly hinders the development and implementation of effective leakage control measures. Even for perturbation, which is the most popular and straightforward privacy-preserving technique, determining the minimal noise needed to meet required security parameters remains an unresolved issue for most practical algorithms (Sridhar et al., 2024). PAC privacy (Xiao & Devadas, 2023) addresses the fundamental problem of quantifying the relationship among data entropy, disclosure, reconstruction difficulty, and

overcoming challenges in classic security and privacy regimes for black-box data processing. It allows for automatic security proof and instance-based worst-case analysis, enabling users to monitor data leakage of any black-box processing mechanism with confidence through simulation. Furthermore, PAC Privacy also guarantees simple composition bounds, and the automatic analysis framework can be implemented online to analyze composite PAC Privacy loss, even with correlated randomness. In terms of utility, different from DP, the necessary perturbation for PAC Privacy is not lower bounded by $\Theta(\sqrt{d})$ for a $d$-dimensional release; instead the bound can be as low as $O(1)$ for many practical tasks, contrasting with the input-independent worst-case information-theoretic lower bound.

**Definition 3.1.** *($(\delta, \rho, D)$ PAC Privacy). For a data processing mechanism $\mathcal{M}$, given some data distribution $D$, a measure function $\rho(.,.)$, and a finite set $X^*$, we say $\mathcal{M}$ satisfies $(\delta, \rho, D)$-PAC Privacy if the following experiment is impossible: A user generates data $X$ from distribution $D$ and sends $\mathcal{M}(\mathcal{X})$ to an adversary. The adversary who knows $D$ and $\mathcal{M}$ is asked to return an estimation $\tilde{X} \in X^*$ on $X$ such that with posterior success probability at least $(1 - \delta)$, $\rho(\tilde{X}, X) = 1$.*

**Definition 3.2.** *($(\Delta_f \delta, \rho, \mathsf{D})$ PAC Advantage Privacy) Equivalantly, $\mathcal{M}$ could be defined as $(\Delta_f \delta, \rho, \mathsf{D})$ PAC Advantage Privacy if the posterior advantage measured in $f$-divergence $\mathcal{D}_f$ satisfies*

$$\Delta_f \delta = \mathcal{D}_f(\mathbf{1}_\delta \| \mathbf{1}_{\delta_o^\rho}) = \delta_o^\rho f(\frac{\delta}{\delta_o^\rho}) + (1 - \delta_o^\rho) f(\frac{1 - \delta}{1 - \delta_o^\rho}), \tag{2}$$

*where $(1 - \delta_o^\rho)$ represents the optimal prior success rate of recovering $X$.*

$$1 - \delta_o^\rho = \sup_{\tilde{X} \in \mathcal{X}^*} \Pr_{X \sim \mathsf{D}} \left( \rho(\tilde{X}, X) = 1 \right), \tag{3}$$

*and $\mathbf{1}_\delta$ and $\mathbf{1}_{\delta_o^\rho}$ represent two Bernoulli distributions of parameters $\delta$ and $\delta_o^\rho$, respectively.*

The definition above depicts the reconstruction hardness for the attackers to recover the private data distribution $\mathcal{M}(\mathcal{X})$. With a lower bound probability $(1 - \delta)$, the measure function $\rho(.,.)$ cannot distinguish the recovery data from the original data. However, the limitation of the naive definition is that the prior distribution of the public dataset is unknown, resulting in the failure of adversarial inferences.

**Definition 3.3.** *(Mutual Information). For two random variables $x$ and $w$ in some joint distribution, the mutual information $MI(x; w)$ is defined as*

$$MI(x, w) = \mathcal{H}(x) - \mathcal{H}(x|w) = \mathcal{D}_{KL}(P_{x,w} \| P_x \otimes P_w), \tag{4}$$

*where $\mathcal{D}_{KL}$ denotes the KL divergence.*

**Theorem 3.1.** *For any selected $f$-divergence $\mathcal{D}_f$, a mechanism $\mathcal{M} : \mathcal{X}^* \to \mathcal{Y}$ satisfies $(\Delta_f \delta, \rho, \mathsf{D})$ PAC Advantage Privacy if*

$$\Delta_f \delta = \mathcal{D}_f(\mathbf{1}_\delta \| \mathbf{1}_{\delta_o^\rho}) \leq \inf_{\mathsf{P}_W} \mathcal{D}_f(\mathsf{P}_{X, \mathcal{M}(X)} \| \mathsf{P}_X \otimes \mathsf{P}_W). \tag{5}$$

*In particular, when we select $\mathcal{D}_f$ to be the KL-divergence and $\mathcal{P}_W = \mathcal{P}_{\mathcal{M}(X)}$, $\mathcal{M}$ satisfies $(\Delta_{KL} \delta, \rho, \mathsf{D})$ PAC Advantage Privacy where*

$$\Delta_{KL} \delta = \mathcal{D}_{KL}(\mathbf{1}_\delta \| \mathbf{1}_{\delta_o^\rho}) \leq MI(X; \mathcal{M}(X)). \tag{6}$$

*Proof.* See Appendix A. □

In summary, $\mathcal{D}_f(\mathbf{1}_\delta \| \mathbf{1}_{\delta_o^\rho})$ quantifies the divergence between optimal a priori and posterior reconstruction, effectively measuring the difficulty of inference. A higher value of $\mathcal{D}_f(\mathbf{1}_\delta \| \mathbf{1}_{\delta_o^\rho})$ signifies greater privacy leakage. Moreover, since $MI(X; \mathcal{M}(X))$ provides an upper bound for $\mathcal{D}_f$, a lower value of $MI(X; \mathcal{M}(X))$ indicates stronger privacy protection. Thus, theorem 3.1 establishes a general method for linking the difficulty of arbitrary inference to the well-known concept of mutual information. With theorem 3.1, the goal of PAC privacy is explicit: determining the bound $MI(X; \mathcal{M}(X))$ with high confidence.

### 3.3 Conditional Diffusion Models

Dhariwal & Nichol (2021) proposed a diffusion model that is enhanced by classifier guidance; it has been shown to outperform existing generative models. By using true labels of datasets, it is possible to train a classifier on noisy images $x_t$ at various timesteps $p_\phi(y|x_t, t)$ , and then use this classifier to guide the reverse sampling process $\nabla_{x_t} \log p_\phi(y|x_t, t)$. What begins as an unconditional reverse noising process is thus transformed into a conditional one, where the generation is directed to produce specific outcomes based on the given labels

$$p_{\theta,\phi}(x_t|x_{t+1}, y) = Z p_\theta(x_t|x_{t+1}) p_\phi(y|x_t) \tag{7}$$

Where $Z$ is a normalizing constant. According to unconditional reverse process, which predicts timestep $x_t$ from $x_{t-1}$ leveraging Gaussian distribution, we have

$$p_\theta(x_t|x_{t+1}) \sim \mathcal{N}(\mu, \Sigma) \tag{8}$$

$$\log p_\theta(x_t|x_{t+1}) = -\frac{1}{2}(x_t - \mu)^T \Sigma^{-1}(x_t - \mu) + C \tag{9}$$

If we assume that $\log_\phi p(y|x_t)$ has low curvature compared to $\Sigma^{-1}$, then we can approximate $\log p_\phi(y|x_t)$ using a Taylor expansion around $x_t = \mu$ as

$$\begin{aligned}\log p_\phi(y|x_t) &\approx \log p_\phi(y|x_t)|_{x_t=\mu} + (x_t - \mu)\nabla_{x_t} \log p_\phi(y|x_t)|_{x_t=\mu} \\ &= (x_t - \mu)g + C_1\end{aligned} \tag{10}$$

where $g$ is the gradient of classifier $g = \nabla_{x_t} \log p_\phi(y|x_t)|_{x=\mu}$ and $C_1$ is the constant.

Therefore, combing Eqn. 9 and 10 gives us

$$\log(p_\theta(x_t|x_{t+1}) p_\phi(y|x_t)) \sim \mathcal{N}(\mu + \Sigma g, \Sigma) \tag{11}$$

The investigation has led to the conclusion that the conditional transition operator can be closely estimated using a Gaussian. And the Gaussian resembles the unconditional transition operator, with the distinction that its mean is adjusted by the product of the covariance matrix, $\Sigma$, and the vector $g$. This methodology allows for the generation of high-quality, targeted synthetic data.

## 4 Methods

We introduce the PAC Privacy Preserving Diffusion Model (P3DM), which aims to safeguard privacy for specific attributes, building upon the foundation set by DPGEN (Chen et al., 2022). First, DPGEN injects Gaussian noise with $\tilde{x}_i = x_i + z_i$ where $z_i$ is sampled from $\mathcal{N}(0, \sigma^2 I)$ given a sensitive dataset $\{x_i : i = 1, 2, ..., m\}_{i=1}^m$, which serves as reconstructing images with noise conditional score networks (Song et al., 2020) leveraged in DPGEN. Second, after obtaining noisy images, DPGEN utilizes the random response method as follows, to privatize the training data.:

$$Pr[H(\tilde{x}_i) = w] = \begin{cases} \dfrac{e^\epsilon}{e^\epsilon + k - 1}, & w = x_i \\ \dfrac{1}{e^\epsilon + k - 1}, & w = x_i' \in X \setminus x_i \end{cases} \tag{12}$$

In the equation, $X = \{x_j : max(\tilde{x}_i - x_j)/\sigma_j \leq \beta, j \in [m]\}$ (where "max" is over the dimensions of $\tilde{x}_i - x_j$), $|X| = k \geq 2$, where the hyperparameter $k$ denotes the number of selected candidates from $m$ training images, and the sampling probability for each image is given by

$$p(x_i) = \exp(-d_\infty(x_i, \tilde{x}_i, \sigma_i)) / \sum_{a=1}^m \exp(-d_\infty(x_i, \tilde{x}_i, \sigma_a)) \tag{13}$$

In other words, the mechanism $H(\cdot)$ consists of 2 steps:

1. Sampling $k$ image candidates from $\{x_i\}_{i=1}^m$, with the sampling probability for each image from Eqn. 13, to construct set $X$.
2. Privatize images in $X$ with RR from Eqn. 12.

In general, DPGEN privatize images via mechanism $\mathcal{M}$ where $\mathcal{M}(x_i) = (H \circ f)$. With the method, $\tilde{x}_i$ is designed to point to one of its $k$ nearest neighbors with certain probability $Pr[H(\tilde{x}_i) = x_i^r]$, which is designed to enforce $\epsilon$-differential privacy by privatizing the recovery direction $d = (\tilde{x}_i - x_i^r)/\sigma_i^2$. Following perturbation, DPGEN learns an energy score function:

$$l(\theta, \sigma) = \frac{1}{2}\mathbb{E}_{p(x)}\mathbb{E}_{\tilde{x} \sim \mathcal{N}(x,\sigma^2)} \left[ \|d - \nabla_x \log q_\theta(\tilde{x})\|^2 \right] \tag{14}$$

Subsequently, images are synthesized via the Langevin MCMC process.

Since the Gaussian noise is added uniformly to every dataset samples, $Pr[H(\tilde{x}_i) = w]$ and the set $X$ in the RR mechanism do not depend on Gaussian noise $Z$. Therefore, we can reverse the order of Gaussian noise addition and the RR mechanism in Algorithm 2.

While DPGEN asserts compliance with stringent $\epsilon$-differential privacy, recent findings Dockhorn et al. (2022) indicate that DPGEN is, in fact, data-dependent. The RR mechanism $M(d)$ in DPGEN holds validity only within the perturbed dataset. Specifically, if an element $z$ belongs to the output set $O$ but not to the perturbed dataset $d$, then $Pr[M(d) = O] = 0$, which contravenes the differential privacy definition.

## 4.1 Conditional Private Langevin Sampling

We introduce a method of conditional private guidance within the Langevin sampling algorithm, detailed in Algorithm 1. This method is designed to protect specific attributes within the original datasets against adversarial attacks.

Prior to commencing the sampling iterations, it is essential to obtain class labels from a balanced attribute of the dataset. The necessity for a dataset attribute that possesses an approximately equal number of negative and positive samples is crucial for training classifiers to achieve exceptional performance. Subsequently, we select a random label $y_i$ from $y$ and fix it to initialize $x_0$ with a predetermined distribution, such as the standard normal distribution.

Drawing on conditional image generation (Dhariwal & Nichol, 2021; Batzolis et al., 2021; Ho & Salimans, 2022), we adapt the model from the vanilla Langevin dynamic samplings with the selected attributes with conditional guidance

$$k\Sigma_\theta(x_{t-1})\nabla_{x_{t-1}} \log c_\theta(y_n|x_{t-1}) \tag{15}$$

where $k$ is the gradient scale that can be tuned according to the performance of the model, $\Sigma_\theta(x_{t-1})$ is the covariance matrix from reverse process Eqn. 8. In Eqn. 15, the term $\nabla_{x_{t-1}} \log c_\theta(y_n|x_{t-1})$ directs the Langevin sampling process toward a specific class label $y_n$, which is sampled prior to the inference.

This private guidance during the inference phase ensures that the synthesized images are protected from privacy breaches related to designated attributes. At its core, this method involves intentionally modifying certain generated trajectories by randomly perturbing the image label $y_n$, which is then used by a pretrained classifier to guide the image generation process. This strategy is aimed at diverting certain image attributes that we wish to protect. For instance, consider a scenario where an original dataset image depicting Celebrity A wearing eyewear, a known trait of the celebrity. If our goal is to privatize the attribute of "wearing glasses," private classifier guidance can effectively achieve this. When sampling occurs under the influence of this guidance, the attributes are likely to vary, creating a chance that the resulting synthesized image based on Celebrity A might be rendered without glasses. This modification effectively protects the attribute of 'glasses' from being a consistent element in the generated depictions of Celebrity A. Consequently, the images generated with this approach offer a higher degree of privacy compared to those produced by the original DPGEN method.

---

**Algorithm 1** PAC-Private Conditional Guidance Langevin Dynamics Sampling

---

**Require:** class labels $y = \{y_1, y_2, ...y_n\}$ from one of balanced dataset attributes;
    gradient scale k; $\{\delta_i\}_{i=1}^L$, $\epsilon$, $T$;
    pretrained scorenet model $s_\theta$;
    pretrained classifier model on noisy images $c_\theta$
 1: Randomly sample $y_n$ from $y$
 2: Initialize $x_0$
 3: **for** $i$ from 1 to $L$ **do**
 4:    stepsize $\alpha_i \leftarrow \delta_i^2/\delta_L^2$
 5:    **for** $t$ from 1 to $T$ **do**
 6:      Sample noise $z_t \sim \mathcal{N}(0, I)$
 7:      $x_t \leftarrow x_{t-1} + \frac{\alpha_i}{2}s_\theta(x_{t-1}, \delta_i) + \sqrt{\alpha_i}z_t + k\Sigma_\theta(x_{t-1})\nabla_{x_{t-1}}\log c_\theta(y_n|x_{t-1})$
 8:    **end for**
 9:    $x_0 \leftarrow x_T$
10: **end for**
11: **Return** $x_T$

---

**Algorithm 2** PAC-Private Adapted Randomized Response Algorithm

---

**Require:** a sensitive dataset $\{x_i : i = 1, 2, ..., m\}_{i=1}^m$; sample number $k$
 1: Sampling $k$ image candidates from $\{x_i\}_{i=1}^m$, with the sampling probability for each image Eqn. 13, to construct set X $\leftarrow \{x_j : max(\tilde{x}_i - x_j)/\sigma_j \leq \beta, j \in [m]\}$
 2: Privatize images in $X$ with RR from Eqn. 12 and obtain $H(x_i)$
 3: $M(\tilde{x}_i) = H(x_i) + z_i$; $z_i \sim \mathcal{N}(0, \sigma^2 I)$
 4: **Return** $M(\tilde{x}_i)$

---

## 4.2 Privacy Metrics

Most privacy-preserving generative models prioritize assessing the utility of images for downstream tasks, yet often overlook the crucial metric of the models' own privacy. To bridge this gap in evaluating privacy extent, we have developed a novel algorithm that computes a privacy score for the models obtained. As per Algorithm 3, we commence by preparing images $x_i$ generated from Algorithm 1, the original dataset images $x_k^G$, and alongside the classifier model which trained separately with clean images (different from the classifier from Algorithm 1). Subsequently, we process all synthesized images through InceptionV3 (Szegedy et al., 2016).

After obtaining the feature vector output from InceptionV3, we locate the feature vector of a ground truth image that has the smallest L2 distance to that of the synthesized image, effectively finding the nearest ground truth neighbor. We then test whether a pretrained classifier model can differentiate these two images. Inability of the classifier to distinguish between the two indicates that the specific attributes, even in images most similar to the original, are well-protected. Thus, our method successfully preserves privacy for specific attributes. Finally, we compute the average probability of incorrect classification by the pretrained model to establish our privacy score. The greater the privacy score a model achieves, the more robust its privacy protection is deemed to be.

## 4.3 PAC Mutual Information Automatic Control

**Theorem 4.1.** *When the mutual information $MI(X; M(X))$ is insufficient to ensure PAC privacy, additional Gaussian noise $B \sim \mathcal{N}(0, \Sigma_B)$ can be introduced to yield a reduced mutual information $MI(X; M(X) + B)$ (Xiao & Devadas, 2023), such that $MI(X; M(X) + B)$ satisfies*

$$\mathsf{MI}(X; \mathcal{M}(X) + \boldsymbol{B}) \leq \frac{1}{2} \cdot \log det\big(\boldsymbol{I}_d + \Sigma_{\mathcal{M}(X)} \cdot \Sigma_{\boldsymbol{B}}^{-1}\big). \tag{16}$$

---

**Algorithm 3** Privacy Score

---

**Require:** images $x_i$ generated by algorithm 1;
    ground truth images $x_k^G$;
    sample number $n$;
    pretrained InceptionV3 model $I_\theta$;
    pretrained classifier model $c_\theta$.
1: private score $s \leftarrow 0$
2: **for** $x_i$ from $x_0$ to $x_n$ **do**
3:     find $argmin_k||I_\theta(x_i), I_\theta(x_k^G)||_2$
4:     **if** $c_\theta(x_i) \neq c_\theta(x_k^G)$ **then**
5:         $s \leftarrow s + 1$
6:     **end if**
7: **end for**
8: **Return** $s/n$

---

*Proof.* See Appendix B. $\qquad\square$

Based on Theorem 4.1, PAC privacy provides a $1 - \gamma$ confidence in the noise determination for the deterministic mechanism $M$ shown in Algorithm 4, to compute the Gaussian noise covariance $\Sigma_B$, ensuring the upper bound $MI(X; M(X) + B) \leq \nu + \beta$. In practical terms, this means we can evaluate the privacy extent of the model by comparing the mean norm $\mathbb{E}||B||_2$ under the same mutual information upper bound level $\nu + \beta$. A smaller $\mathbb{E}||B||_2$ implies a smaller covariance matrix $\Sigma_B$ and lesser noise addition on $M(X)$, which indicates $X$ and $M(X)$ have less mutual information. Therefore, less noise addition signifies a higher privacy protection of the model $M$.

### 4.4   PAC Privacy Proof of Our Model

Since the randomized response method adds Gaussian noise $Z$ in the random response mechanism, it can be proven to be PAC private using Theorem 3.1 and Theorem 4.1 in the paper:

- Algorithm 2 can be combined and written as $H(X) + \boldsymbol{B}$, where $H(X)$ represents the RR mechanism and $\boldsymbol{B}$ denotes Gaussian noise with $\boldsymbol{B} \sim \mathcal{N}(0, \Sigma_B)$.

- Applying Theorem 4.1, we can yield a satisfied upper bound for the whole process $MI(X; \mathcal{M}(X)) \leq \frac{1}{2} \cdot \log \det\left(\boldsymbol{I}_d + \Sigma_{\mathcal{M}(X)} \cdot \Sigma_{\boldsymbol{B}}^{-1}\right)$, where $\mathcal{M}(X) = H(X) + \boldsymbol{B}$.

- Hence, $MI(X; \mathcal{M}(X))$ can be used to produce a (loose) upper bound of PAC Privacy via Theorem 3.1.

What's more, Algorithm 1 is a heuristic method that further enhances privacy. We calculate noise $B$ to measure its privacy strength, achieving the best performance as shown in Table 2. Of independent interest to ensure the PAC privacy of Algorithm 1 itself, we simply need to add noise $B$ to the final $x_T$ in Algorithm 1.

## 5   Experiments

### 5.1   Datasets

Our experiments were carried out using the CelebA (Liu et al., 2015) datasets. We specifically targeted attributes that have a balanced distribution of positive and negative samples, as noted in Rudd et al. (2016), to facilitate the training of classifiers. Consequently, we selected attributes like gender and smile from the CelebA dataset (referred to as CelebA-gender and CelebA-smile, respectively) for sampling in Algorithm 1. All the images used in these experiments were of the resolution $64 \times 64$.

---

**Algorithm 4** $(1 - \gamma)$-Confidence Noise Determination of Deterministic Mechanism

---

**Require:** Diffusion-Privacy model $M$, data distribution $D$, sampling complexity $m$, security parameter $c$, and mutual information quantities $\nu$ and $\beta$.

1: Train $M$ model with data $X$.
2: **for** $k = 1, 2, \ldots, m$ **do**
3:      sample images $y^{(k)} = M(X^{(k)})$.
4: **end for**
5: Calculate empirical mean $\hat{\mu} = \frac{1}{m} \sum_{k=1}^{m} y^{(k)}$ and the empirical covariance estimation $\hat{\Sigma} = \frac{1}{m} \sum_{k=1}^{m} (y^{(k)} - \hat{\mu})(y^{(k)} - \hat{\mu})^T$.
6: Apply singular value decomposition (SVD) on $\hat{\Sigma}$ and obtain the decomposition as $\hat{\Sigma} = \hat{U} \hat{\Lambda} \hat{U}^T$, where $\hat{\Lambda}$ is the diagonal matrix of eigenvalues $\lambda_1 \geq \lambda_2 \geq \ldots \geq \lambda_d$.
7: Determine the maximal index $j_0 = \arg\max_j \lambda_j$ for those $\lambda_j > c$.
8: **if** $\min_{1 \leq j \leq j_0, 1 \leq d}(\lambda_j - \hat{\lambda}_j) > r\sqrt{d/c + 2c}$ then **then**
9:      **for** $j = 1, 2, \ldots, d$ **do**
10:          Determine the $j$-th element of a diagonal matrix $A_B$ as

$$\lambda_{B,j} = \frac{2\nu}{\sqrt{\lambda_j + 10c\nu/\beta} \cdot \left(\sum_{j=1}^{d} \sqrt{\lambda_j + 10c\nu/\beta}\right)}$$

11:      **end for**
12:      Determine the Gaussian noise covariance as $\Sigma_B = \hat{U} A_B \hat{U}^T$.
13: **else**
14:      Determine the Gaussian noise covariance as $\Sigma_B = \left(\sum_{j=1}^{d} \lambda_j + dc/(2\nu)\right) \cdot I_d$.
15: **end if**
16: **Return** Gaussian covariance matrix $\Sigma_B$.

---

## 5.2 Baselines

In our study, we consider DPGEN (Chen et al., 2022), DPDM (Dockhorn et al., 2022) and DP-MEPF (Harder et al., 2023) as baseline methods. Both of these approaches excel in synthesizing images under differential privacy (DP) constraints, and they stand out for their exceptional sample quality in comparison to other DP generative models. These models serve as important benchmarks against which we evaluate the performance and efficacy of our proposed method.

## 5.3 Evaluation Metrics

In our evaluation process, we validate the capability of our PAC Privacy Preserving Diffusion Model to generate high-resolution images using two key metrics: (1) the Frechet Inception Distance (Heusel et al., 2017) and (2) the Inception Score (Salimans et al., 2016). These metrics are widely recognized and utilized in the field of generative models to assess the visual fidelity of the images they produce.

Additionally, to demonstrate our model's effectiveness in preventing privacy leakage, we employ our unique privacy metrics as outlined in Algorithm 3. We compare the mean norm of the Gaussian noise $\mathbb{E}||B||_2$ as detailed in Sec. 4.3. For our experiments, we have chosen the hyperparameters $\nu = \beta = 0.5$ and $\gamma = 0.01$. This approach allows us to comprehensively assess not just the quality of the images generated, but also the strength of privacy protection our model offers.

## 5.4 Empirical Results And Analysis

It is important to note that the $\xi$ presented in our P3DM model within the tables below is merely a hyperparameter derived from the Randomized Response (RR) as indicated in Eqn. 12. This $\xi$ should not be confused with the $\varepsilon$ from $\varepsilon$-Differential Privacy (DP), as our model uses PAC privacy rather than $\varepsilon$-DP.

Table 1: Perceptual and privacy score comparisons on CelebA with image resolution $64 \times 64$. In our model, we train with data on a specific label respectively, and Model-Gender means our model is trained with CelebA-Gender dataset. $\epsilon$ represents the DP parameter of baselines, while $\xi$ indicates a hyperparameter derived from the Randomized Response (RR). The same applies to the following.

| $\varepsilon$ or $\xi$ | P3DM-Gender | | P3DM-Smile | | DPGEN | | DPDM | | DP-MEPF | |
|---|---|---|---|---|---|---|---|---|---|---|
| | FID ↓ | Privacy Score ↑ | FID ↓ | Privacy Score ↑ | FID ↓ | Privacy Score ↑ | FID ↓ | Privacy Score ↑ | FID ↓ | Privacy Score ↑ |
| 1 | 175±2.5 | 0.6±0.05 | 170±1.85 | **0.63±0.08** | | | | | **67.5±1.45** | 0.48±0.02 |
| 5 | 125.8±3.6 | 0.55±0.082 | 107.85±4.7 | **0.6±0.075** | 155±1.5 | 0.46±0.076 | 170±2 | 0.57±0.06 | **61.2±1.8** | 0.45±0.03 |
| 10 | 44.82±4.48 | 0.5±0.05 | **37.96±1.85** | **0.56±0.045** | 39.16±0.68 | 0.4±0.03 | 117±1.5 | 0.47±0.04 | 57.5±1.2 | 0.4±0.04 |
| 15 | 40±0.5 | 0.45±0.05 | **37.9±0.6** | **0.5±0.025** | 38.5±0.73 | 0.4±0.06 | 113±1.2 | 0.35±0.05 | 55.8±1.4 | 0.38±0.02 |
| ∞ | 34.64±2.18 | 0.4±0.04 | 36.03±1.32 | **0.48±0.055** | **34.4±1.8** | 0.376±0.03 | 110±1.2 | 0.3±0.06 | 52±2.3 | 0.34±0.025 |

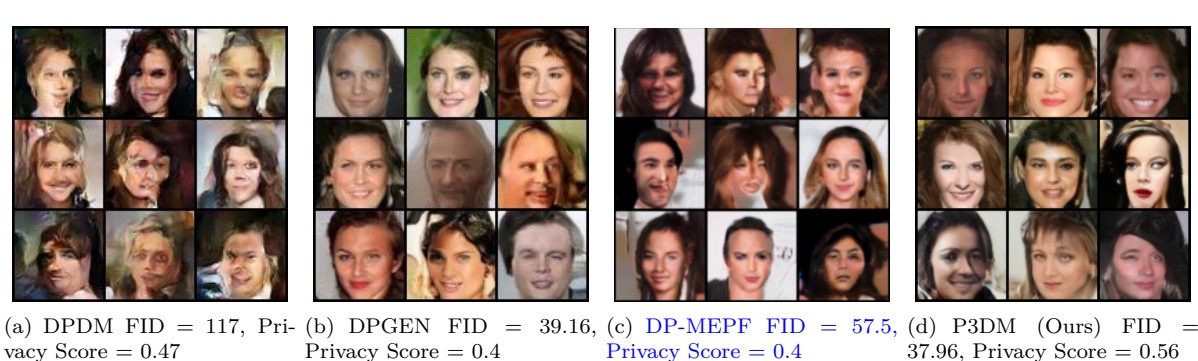

(a) DPDM FID = 117, Privacy Score = 0.47

(b) DPGEN FID = 39.16, Privacy Score = 0.4

(c) DP-MEPF FID = 57.5, Privacy Score = 0.4

(d) P3DM (Ours) FID = 37.96, Privacy Score = 0.56

Figure 1: CelebA images generated from DPDM, DPGEN and our model from left to right with image resolution $64 \times 64$.

We will later illustrate how our model assures privacy through the automatic control of mutual information for a PAC privacy guarantee in this section.

Table 1 and Fig. 3 details the evaluation results of image visual quality and privacy score on the CelebA dataset with a resolution of $64 \times 64$, where each datapoint in the figure, or each entry in the table, consists of mean and standard deviation from 3 experimental results with the same epsilon and different random seeds. By examining the table, we can see that our model achieves image quality comparable to the state-of-the-art model DPGEN (Chen et al., 2022), a conclusion also supported by Fig.1.

Additionally, from Fig. 3, our model registers the highest performance in privacy score when having similar FIDs, signaling an enhancement in our model's ability to preserve privacy without substantially impacting image utility. The assertion is further supported by Fig. 2, wherein the CelebA dataset is filtered based on the 'smile' attribute. Even upon querying the nearest images of P3DM samples, we can distinctly observe that while all P3DM samples exhibit the absence of a smile, the nearest neighbors predominantly display smiling faces. This highlights that, in contrast to other models, the closest images between the generated dataset and the ground truth datasets are notably similar in terms of the 'smile' expression, while the images generated by our model demonstrate distinctive features. Hence, the P3DM model effectively conceals the 'smile' attribute.

Furthermore, in Table 2, we compute the multivariate Gaussian matrix $B$ with a $1 - \gamma$ noise determination for the deterministic mechanism $M$ followed the work by Xiao & Devadas (2023). From this, we show that under the same confidence level of $1 - \gamma = 0.99$ to ensure $MI(X; M(X) + B) \leq 1$, the PAC Privacy Preserving Diffusion Model exhibits the smallest norm on $\mathbb{E}||B||_2$. This demonstrates that data processed by our model retains the least mutual information with the original dataset, thus affirming our model's superior performance in privacy protection.

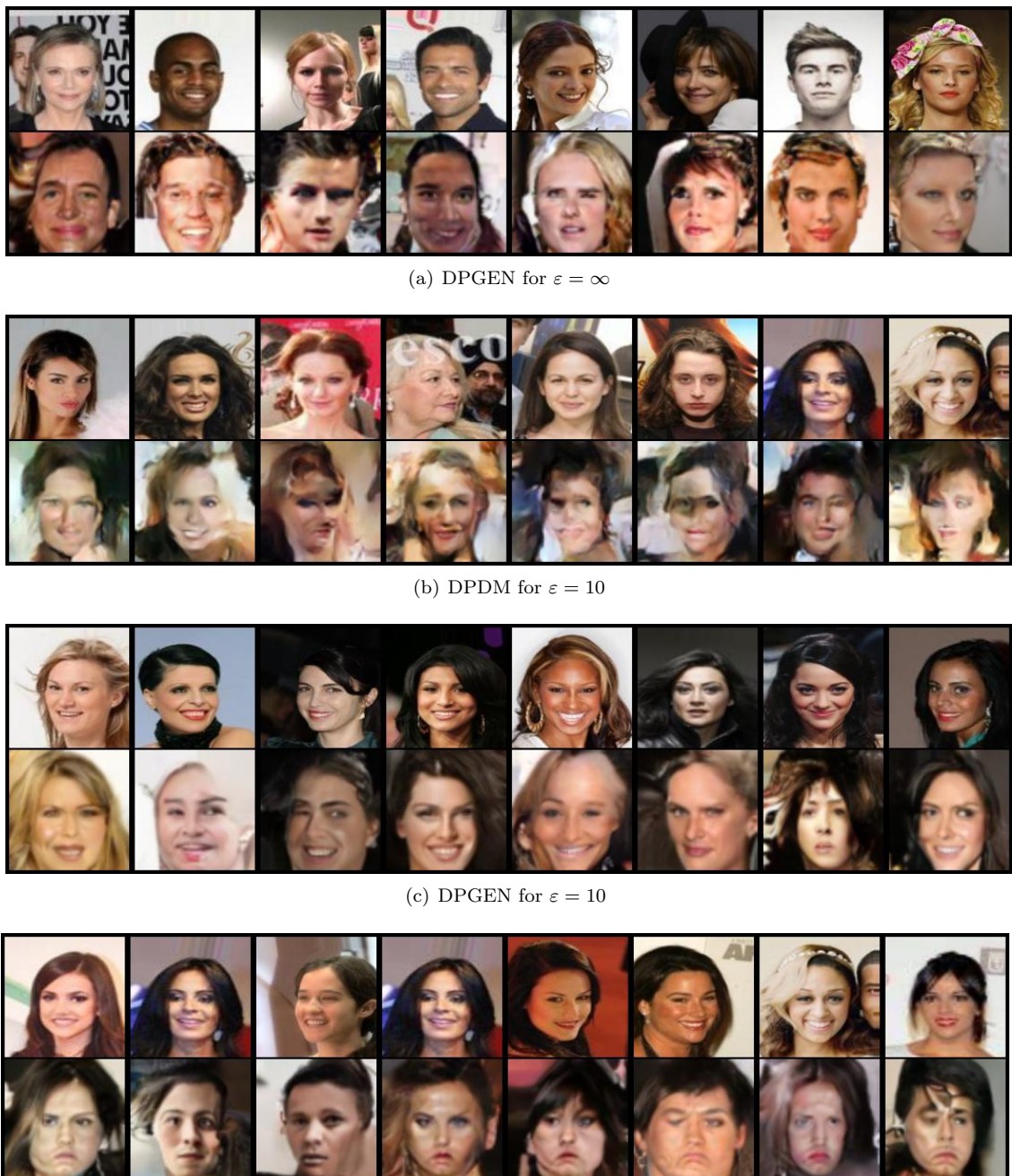

(a) DPGEN for $\varepsilon = \infty$

(b) DPDM for $\varepsilon = 10$

(c) DPGEN for $\varepsilon = 10$

(d) P3DM-Smile (Ours) for $\xi = 10$

Figure 2: Generated images (the second row) and their nearest neighbors measured by the $l_2$ distance between images from CelebA-smile dataset (the first row), with image resolution $64 \times 64$.

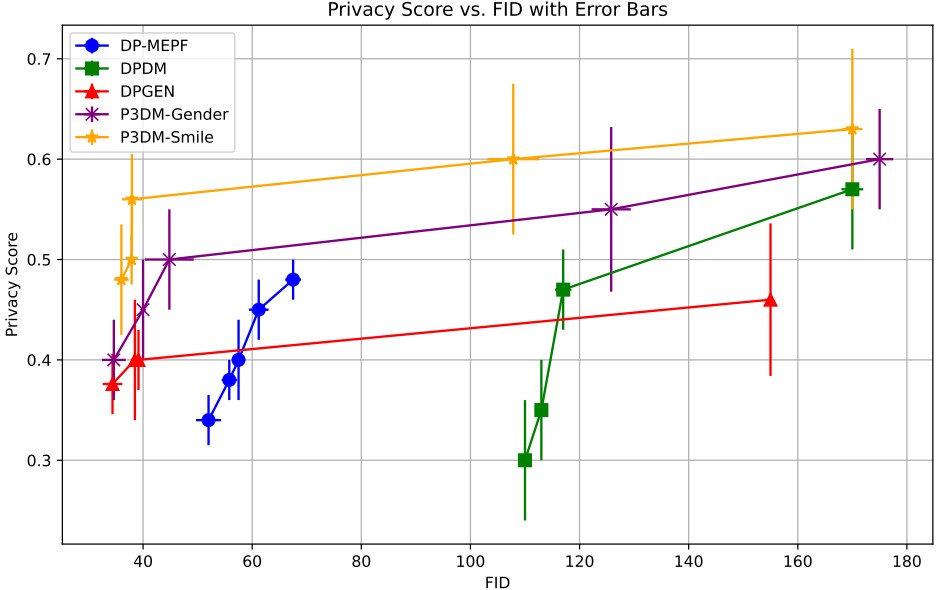

Figure 3: **Privacy score and FID curve of all datapoints from different models.** Each datapoint in the figure consists of mean and standard deviation from 3 experimental results with the same epsilon and different random seeds. Top-right corner is preferred. The curve from our method, pushes the frontier to the upper-right over DPGEN (Chen et al., 2022), DPDM (Dockhorn et al., 2022) and DP-MEPF (Harder et al., 2023).

## 6 Limitations

The dataset attributes analyzed in our study predominantly consist of features with balanced positive and negative samples. Attributes characterized by imbalanced distributions remain underexplored. Furthermore, it should be noted that not all datasets include annotated attributes. Therefore, utilizing image-to-text models such as CLIP (Radford et al., 2021) may prove beneficial for the generation of new attribute labels. Lastly, our model faces challenges in synthesizing higher-resolution privacy-preserving images (e.g., 256x256 pixels) and handling complex datasets such as the CUB dataset (Wah et al., 2011)).

## 7 Conclusion

We introduce the PAC Privacy Preserving Diffusion Model (P3DM), which incorporates conditional private classifier

Table 2: $\mathbb{E}||B||_2$ to ensure $MI(X; M(X) + B) \leq 1$ on CelebA dataset

| Methods | $\varepsilon$ or $\xi$ | $\mathbb{E}||B||_2 \downarrow$ |
|---|---|---|
| **P3DM-Gender** | 5 | 283.03±2.25 |
| **P3DM-Smile** | 5 | **280.60±2.57** |
| DPGEN | 5 | 281.3±2.78 |
| DP-MEPF | 5 | 325.6±3.62 |
| **P3DM-Gender** | 10 | 328.80±1.24 |
| **P3DM-Smile** | 10 | **327.98±1.45** |
| DPGEN | 10 | 329.48±1.6 |
| DPDM | 10 | 335.83±1.21 |
| DP-MEPF | 10 | 330.75±1.33 |
| **P3DM-Gender** | ∞ | 332.87±1.15 |
| **P3DM-Smile** | ∞ | **331.67±1.3** |
| DPGEN | ∞ | 332.02±1.32 |
| DPDM | ∞ | 335.83±1.56 |
| DP-MEPF | ∞ | 333.48±1.79 |

guidance into the Langevin Sampling process to selectively privatize image features. In addition, we have developed and implemented a unique metric for evaluating privacy. This metric involves comparing a generated image with its nearest counterpart in the dataset to assess whether a pretrained classifier can differentiate between the two. Furthermore, we calculate the necessary additional noise $B$ to ensure PAC privacy and benchmark the noise magnitude against other models. Our thorough empirical and theoretical testing confirms that our model surpasses current state-of-the-art private generative models in terms of privacy protection while maintaining comparable image quality.

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

# A    Proof of Theorem 3.1

*Proof.* To start, we need the following lemmas.

**Lemma A.1.** *Given any $f$-divergence $\mathcal{D}_f(\cdot\|\cdot)$, and three Bernoulli distributions $\mathbf{1}_a$, $\mathbf{1}_b$ and $\mathbf{1}_c$ of parameters $a$, $b$ and $c$, respectively, where $0 \le a \le b \le c \le 1$. Then, $\mathcal{D}_f(\mathbf{1}_a\|\mathbf{1}_b) \le \mathcal{D}_f(\mathbf{1}_a\|\mathbf{1}_c)$.*

*Proof.* By the definition, $g(x) = \mathcal{D}_f(\mathbf{1}_a\|\mathbf{1}_x) = xf(\frac{a}{x}) + (1-x)f(\frac{1-a}{1-x})$ and we want to show $g(x)$ is non-decreasing for $x \ge a$. With some calculation, $g'(x) = \left(f(\frac{a}{x}) - \frac{a}{x}f'(\frac{a}{x})\right) - \left(f(\frac{1-a}{1-x}) - \frac{1-a}{1-x}f'(\frac{1-a}{1-x})\right)$. It is noted that $\frac{a}{x} \le \frac{1-a}{1-x}$ for $x \ge a$. Thus, to show $g'(x) \ge 0$ for $x \ge a$, it suffices to show $t(y) = f(y) - yf'(y)$ is non-increasing with respect to $y \in [0,1]$. On the other hand, $t'(y) = f'(y) - f'(y) - yf''(y) \le 0$ due to the convex assumption of $f$. Therefore, the claim holds. $\square$                               $\square$

**Lemma A.2** (Data Processing Inequality Sason & Verdú (2016)). *Consider a channel that produces $Z$ given $Y$ based on the law described as a conditional distribution $\mathsf{P}_{Z|Y}$. If $\mathsf{P}_Z$ is the distribution of $Z$ when $Y$ is generated by $\mathsf{P}_Y$, and $\mathsf{Q}_Z$ is the distribution of $Z$ when $Y$ is generated by $\mathsf{Q}_Y$, then for any f-divergence $\mathcal{D}_f$,*

$$\mathcal{D}_f(\mathsf{P}_Z\|\mathsf{Q}_Z) \le \mathcal{D}_f(\mathsf{P}_Y\|\mathsf{Q}_Y).$$

Now, we return to prove Theorem 3.1. First, we have the observation that for a random variable $X' \in \mathcal{X}^*$ in an arbitrary distribution but independent of $X$, $\delta_o^\rho \le \Pr_{X'\perp X}\left(\rho(X', X) = 1\right)$, since $\delta_o^\rho$ is the minimum failure probability achieved by optimal *a priori* estimation. Here, $a \perp b$ represents that $a$ is independent of $b$. Let the indicator be a function that for two random variables $a$ and $b$, $\mathbf{1}(a,b) = 1$ if $\rho(a,b) = 1$, otherwise 0. Apply Lemma A.1 and Lemma A.2, where we view $\mathbf{1}(\cdot,\cdot)$ as a post-processing on $(X, \tilde{X})$ and $(X, X')$, respectively, we have that

$$\mathcal{D}_f\left(\mathbf{1}_\delta\|\mathbf{1}_{\delta_o^\rho}\right) \le \mathcal{D}_f\left(\mathbf{1}(X, \tilde{X})\|\mathbf{1}(X, X')\right) \le \mathcal{D}_f\left(\mathsf{P}_{X,\tilde{X}}\|\mathsf{P}_{X,X'}\right) = \mathcal{D}_f\left(\mathsf{P}_{X,\tilde{X}}\|\mathsf{P}_X \otimes \mathsf{P}_{X'}\right).$$

On the other hand, we know $X \to \mathcal{M}(X) \to \tilde{X}$ forms a Markov chain, where the adversary's estimation $\tilde{X}$ is dependent on observation $\mathcal{M}(X)$. Let the adversary's strategy be some operator $g_{adv}$ where $\tilde{X} = g_{adv}(\mathcal{M}(X))$. Therefore, we can apply the data processing inequality again, where

$$\mathcal{D}_f\left(\mathsf{P}_{X,\tilde{X}}\|\mathsf{P}_{X,X'}\right) \le \mathcal{D}_f\left(\mathsf{P}_{X,\mathcal{M}(X)}\|\mathsf{P}_{X,W}\right) = \mathcal{D}_f\left(\mathsf{P}_{X,\mathcal{M}(X)}\|\mathsf{P}_X \otimes \mathsf{P}_W\right).$$

Here, $X' = g_{adv}(W)$ and $W$ is still independent of $X$. Since the above inequalities hold for arbitrarily distributed $X'$ once it is independent of $X$, $W$ could also be an arbitrary random variable on the same support domain as $\mathcal{M}(X)$ and independent of $X$. Therefore,

$$\mathcal{D}_f\left(\mathbf{1}_\delta\|\mathbf{1}_{\delta_o^\rho}\right) \le \inf_{\mathsf{P}_W} \mathcal{D}_f\left(\mathsf{P}_{X,\mathcal{M}(X)}\|\mathsf{P}_X \otimes \mathsf{P}_W\right) = \inf_{\mathsf{P}_W} \mathcal{D}_f\left(\mathsf{P}_{\mathcal{M}(X)|X}\|\mathsf{P}_W|\mathsf{P}_X\right).$$

Here, we use $|\mathsf{P}_X$ to denote that it is conditional on $X$ in a distribution $\mathsf{P}_X$. In particular, if we select $\mathsf{P}_W$ to be the distribution of $\mathcal{M}(X)$, and take $\mathsf{D}_f$ to be KL-divergence, we have $\mathcal{D}_{KL}\left(\mathbf{1}_\delta\|\mathbf{1}_{\delta_o^\rho}\right) \le \mathsf{MI}(X; \mathcal{M}(X))$.   $\square$

# B    Proof of Theorem 4.1

*Proof.* For $\mathsf{MI}(X; \mathcal{M}(X) + B)$, we have

$\mathsf{MI}(X; \mathcal{M}(X) + B)$

$$= \int \mathcal{D}_{KL}(\mathsf{P}_{\mathcal{M}(X_0)+B}\|\mathsf{P}_B)\mathsf{P}(X = X_0)\, dX_0 - \mathcal{D}_{KL}(\mathsf{P}_{\mathcal{M}(X)+B}\|\mathsf{P}_B)$$

$$= \int \mathcal{D}_{KL}(\mathsf{P}_{\mathcal{M}(X_0)+B}\|\mathsf{P}_B)\mathsf{P}(X = X_0)\, dX_0 - \left(\mathcal{D}_{KL}(\mathsf{P}_{\mathcal{M}(X)+B}\|\mathsf{P}_{Gau(\mathcal{M}(X)+B)}) + \mathcal{D}_{KL}(\mathsf{P}_{Gau(\mathcal{M}(X)+B)}\|\mathsf{P}_B)\right)$$

$$\le \int \mathcal{D}_{KL}(\mathsf{P}_{\mathcal{M}(X_0)+B}\|\mathsf{P}_B)\mathsf{P}(X = X_0)\, dX_0 - \mathcal{D}_{KL}(\mathsf{P}_{Gau(\mathcal{M}(X)+B)}\|\mathsf{P}_B).$$

$$(17)$$

Given the definition of mutual information, we first apply the results of Gaussian approximation Pinsker et al. (1995), where $Gau(A)$ represents a (multivariate) Gaussian variable with the same mean and (co)variance as those of $A$. Then, we drop a negative term to obtain the final inequality of (17). Next, focusing on the integral (first) term in (17),

$$\mathcal{D}_{KL}(\mathsf{P}_{\mathcal{M}(X_0)+B} \| \mathsf{P}_B) = \frac{1}{2} \cdot (\mathcal{M}(X_0))^T \Sigma_{\boldsymbol{B}}^{-1} (\mathcal{M}(X_0)),$$

and therefore

$$\int \mathcal{D}_{KL}(\mathsf{P}_{\mathcal{M}(X_0)+B} \| \mathsf{P}_B) \mathsf{P}(X = X_0) \, dX_0 = \frac{1}{2} \cdot \mathbb{E}_X \left[ (\mathcal{M}(X))^T \Sigma_{\boldsymbol{B}}^{-1} (\mathcal{M}(X)) \right].$$

As for the second term in the last equation of (17), we have the covariance of $\mathcal{M}(X)$ equals $\Sigma_{\mathcal{M}(X)} = \mathbb{E}_X \left[ (\mathcal{M}(X) - \mathbb{E}[\mathcal{M}(X)])(\mathcal{M}(X) - \mathbb{E}[\mathcal{M}(X)])^T \right]$, while the mean $\mu_{\mathcal{M}(X)} = \mathbb{E}[\mathcal{M}(X)]$. The KL divergence between two multivariate Gaussians has a closed form, where $\mathcal{D}_{KL}(\mathsf{P}_{Gau(\mathcal{M}(X)+B)} \| \mathsf{P}_B)$ equals

$$\begin{aligned} \mathcal{D}_{KL}(\mathsf{P}_{Gau(\mathcal{M}(X)+B)} \| \mathsf{P}_B) = \frac{1}{2} \cdot \big( &\mathrm{Trace}(\Sigma_{\mathcal{M}(X)} \cdot \Sigma_{\boldsymbol{B}}^{-1}) \\ &+ \mathbb{E}_X[\mathcal{M}(X)]^T \Sigma_{\boldsymbol{B}}^{-1} \mathbb{E}_X[\mathcal{M}(X)] - \log\det(\boldsymbol{I}_d + \Sigma_{\mathcal{M}(X)} \Sigma_{\boldsymbol{B}}^{-1}) \big). \end{aligned} \tag{18}$$

On the other hand, note that

$$\begin{aligned} &\mathbb{E}_X \left[ (\mathcal{M}(X))^T \Sigma_{\boldsymbol{B}}^{-1} (\mathcal{M}(X)) \right] - \mathbb{E}_X \left[ \mathcal{M}(X) \right]^T \Sigma_{\boldsymbol{B}}^{-1} \mathbb{E}_X \left[ \mathcal{M}(X) \right] - \mathrm{Trace}(\Sigma_{\mathcal{M}(X)} \cdot \Sigma_{\boldsymbol{B}}^{-1}) \\ &= \mathrm{Trace}\big( \mathbb{E}[\mathcal{M}(X) - \mathbb{E}[\mathcal{M}(X)]] \cdot \Sigma_{\boldsymbol{B}}^{-1} \cdot \mathbb{E}[\mathcal{M}(X) - \mathbb{E}[\mathcal{M}(X)]]^T \big) - \mathrm{Trace}\big( \Sigma_{\mathcal{M}(X)} \Sigma_{\boldsymbol{B}}^{-1} \big) \\ &= \mathrm{Trace}\big( \mathbb{E}[\mathcal{M}(X) - \mathbb{E}[\mathcal{M}(X)]] \cdot \mathbb{E}[\mathcal{M}(X) - \mathbb{E}[\mathcal{M}(X)]]^T \cdot \Sigma_{\boldsymbol{B}}^{-1} - \Sigma_{\mathcal{M}(X)} \Sigma_{\boldsymbol{B}}^{-1} \big) \\ &= \mathrm{Trace}\big( \Sigma_{\mathcal{M}(X)} \cdot \Sigma_{\boldsymbol{B}}^{-1} - \Sigma_{\mathcal{M}(X)} \Sigma_{\boldsymbol{B}}^{-1} \big) = 0. \end{aligned} \tag{19}$$

In (19), we use the following facts that for two arbitrary vectors $v_1, v_2 \in \mathbb{R}^d$, $(v_1)^T v_2 = \mathrm{Trace}(v_1 \cdot (v_2)^T)$, and for two arbitrary matrices $A_1, A_2 \in \mathbb{R}^{d \times d}$, $\mathrm{Trace}(A_1 A_2) = \mathrm{Trace}(A_2 A_1)$. Therefore, putting it all together, we have a simplified form of the right hand of (17), where

$$\mathsf{MI}(X; \mathcal{M}(X) + B) \leq \frac{\log\det(\boldsymbol{I}_d + \Sigma_{\mathcal{M}(X)} \cdot \Sigma_{\boldsymbol{B}}^{-1})}{2}. \tag{20}$$

$\square$

