# OpenReview forum: "PAC Privacy Preserving Diffusion Models"
_TMLR — Rejected by TMLR_

### Review · Reviewer_AzXa · 2024-05-08

**Summary Of Contributions:**

This paper modifies a class guided diffusion model to include the newly defined PAC privacy guarantee. This algorithm is then applied to the CelebA dataset generation claims. The authors claim that this new approach is pareto-optimal in a range of image fidelity/privacy score.

**Audience:**

Yes

**Broader Impact Concerns:**

Please ensure all your assumptions are justified. We must be clear what constitutes privacy here, since it is a topic of broad concern.

**Claims And Evidence:**

No

**Requested Changes:**

See the comments above in the weakness section, but in addition please comment on:

* What is the connection between PAC-Privacy and rate distortion theory (i.e., lossy compression)?
* Is there any convincing more convincing reasons you can include to consider PAC privacy over DP?
* $I$ should be reserved for mutual information. Change other uses of $I$ to something else, and use $I$ for mutual information
*  Is there a connection with [1]?
* Several years ago there was some information-theoretic work on generalization and lagevin dynamics [2]. Is there any connection here?
* I suggest you do not use $\epsilon$ for anything other than a DP parameter.


[1] Cuff, Paul, and Lanqing Yu. "Differential privacy as a mutual information constraint." Proceedings of the 2016 ACM SIGSAC Conference on Computer and Communications Security. 2016.

[2] Negrea, Jeffrey, et al. "Information-theoretic generalization bounds for SGLD via data-dependent estimates." Advances in Neural Information Processing Systems 32 (2019).

**Strengths And Weaknesses:**

**Strengths**

* Applies a new and potentially interesting notion of privacy
* Considers an important and timely problem of privacy in generative models

**Weaknesses**
* Since this is a new notion of privacy, the authors need to explain why it is important to study
* What is the advantage of considering it over DP
* PAC Privacy is not introduced until 1/3 of the way into the paper, which is too late in my opinion
* First equation in section 4, which is important to understand is very difficult to parse. Define variables and functions before
* Unsubstantiated/unrigorous claims:
> To illustrate, consider an image that is autonomously generated to depict Celebrity A donning
eyewear, thereby reflecting the known fact that Celebrity A habitually wears glasses. However, when sampling
is conducted with the application of private classifier guidance, the attributes are subject to variation.
Consequently, there is a possibility that Celebrity A may be depicted without eyewear in the resulting
image. This alteration effectively safeguards the attribute of ’glasses’ from being a consistent feature in the
generated representation of Celebrity A. As a result, the images generated under this guidance have a higher
privacy extent when compared to those produced by the original DPGEN.

---

> ### Author Response · Authors · 2024-05-20
> **Thank you for your feedback!**
>
> Thank you for your encouraging comments. We are glad that you found the application of the notion of PAC privacy new and potentially interesting, and considered it to be an important and timely problem of privacy in generative models. Below we address your questions one by one. Besides, we have revised various paragraphs, replenished variables and functions for equations, and polished the claims in the paper according to your comments. All changes are marked in blue in the revision.
>
> **Q1. Connection between PAC-Privacy and rate distortion theory (i.e., lossy compression):**
> * **Connections:** Both approaches utilize mutual information to establish bounds.
> * **Differences:** PAC-Privacy focuses on minimizing the mutual information of the recovered data, particularly for sensitive attributes, whereas Rate Distortion Theory aims to maximize the mutual information of the recovered data. This maximization minimizes information loss by maintaining the rate distortion function's lower bound for the transmission bit rate.
>
> **Q2. More convincing reasons you can include to consider PAC privacy over DP**
> * Empirically, as shown in Figure 3 of our paper, our model achieves the highest privacy scores among models with comparable image quality (measured by Fréchet Inception Distances (FIDs)), indicating an improvement in our model’s capacity to preserve privacy without significantly compromising image utility.
>
> * Theoretically, PAC privacy[3] diverges from traditional Differential Privacy (DP) by focusing on the difficulty of reconstructing data using any measure function, thereby offering broader applicability beyond the adversarial worst-case scenarios considered by DP.
>     * Regarding **utility**, the necessary perturbation magnitude in PAC Privacy is not necessarily constrained by a lower bound of ${\Theta}(\sqrt{d})$ for a $d$-dimensional release. Instead, it could be as low as $(O(1))$ for many practical data processing tasks. This stands in contrast to the input-independent worst-case information-theoretic lower bound.
>     * Furthermore, leveraging PAC privacy comes with a **framework** that autonomously determines the minimal noise addition necessary for effective data protection [3]. This framework simplifies the process of achieving privacy-preserving data handling, making it more accessible for widespread use in the field of data science and beyond.
>
>
>
> **Q3. $I$ should be reserved for mutual information. Change other uses of $I$ to something else, and use $I$ for mutual information**
> * We apologize for the confusion. In this paper, the term "MI" as a whole is an abbreviation for "mutual information". Consequently, the letter "I" in "MI" does not have any standalone significance, unlike the "I" in the identity matrix, where it represents an important mathematical concept.
>
>
>
> **Q4. Several years ago there was some information-theoretic work on generalization and langevin dynamics [2]. Is there any connection here?**
> * **Connections**:
>    1.   Both [2] and our paper apply principles of information theory and incorporate noise addition.
>
> * **Differences**:
>    1.  Reference [2] offers information-theoretic generalization bounds during the training phase, whereas our generalization bounds are applicable during the inference phase of generative models.
>    2.  While [2] conducts empirical comparisons on classification tasks, our model focuses on image generation using diffusion models.
>    3.  The primary objective of [2] is to establish performance-related generalization bounds, whereas our goal is to ensure privacy.
>
> **Q5. Is there a connection with [1]?**
> * **Connections:** Both of PAC privacy[3] and MI-DP[1] apply mutual information to directly measure data privacy
>
> * **Differences:**  As we mentioned in **Q2** above, the core objective of DP is to evaluate the distinguishability between processed and original data, while PAC privacy mainly focuses reconstruction/inference hardness under any selection of the measure function.
>
>
> **Q6. I suggest you do not use $\epsilon$ for anything other than a DP parameter.**
> * We have modified $\epsilon$ with $\xi$ for anything other than a DP parameter.
>
>
>
> [1] Cuff, Paul, and Lanqing Yu. "Differential privacy as a mutual information constraint." Proceedings of the 2016 ACM SIGSAC Conference on Computer and Communications Security. 2016.
>
> [2] Negrea, Jeffrey, et al. "Information-theoretic generalization bounds for SGLD via data-dependent estimates." Advances in Neural Information Processing Systems 32 (2019).
>
> [3] Hanshen Xiao and Srinivas Devadas. Pac privacy: Automatic privacy measurement and control of data processing. In Annual International Cryptology Conference, pp. 611–644. Springer, 2023.

---

> > ### Author Response · Authors · 2024-07-14
> > **Follow-Up Again on Response to Review Comments**
> >
> > Thank you again for your encouraging and insightful comments!  We wanted to follow up again regarding our response to your comments, as it has been a week since our last message. We would greatly appreciate any additional feedback or clarifications you might have at your earliest convenience.
> >
> > Thank you very much for your time and consideration.

---

> ### Author Response · Authors · 2024-05-29
> **Follow-Up on Response to Review Comments**
>
> Thank you for your encouraging and constructive comments. We wanted to follow up regarding our previous response to your comments. We would appreciate any additional feedback and welcome any additional questions you might have.

---

> ### Comment · Reviewer_AzXa · 2024-07-03
> **To Authors**
>
> Dear Authors
>
> I have read the newly added content in your revised paper. Would you please write a comment to me about the questions I asked in the "requested changes" section? I will await your response before I submit my recommendation.

---

> ### Author Response · Authors · 2024-07-04
> **Apology for the Configuration Issue of Comments**
>
> Dear Reviewer AzXa,
>
> We apologize for any inconvenience. Our comments for your questions in the “requested changes” section were not visible due to a configuration issue. This issue has been resolved, and our comments should be visible to you now. Feel free to let us know if there is anything that needs further clarification, and look forward to any follow-up discussion with you!
>
> Best regards,
>
> Authors of P3DM

---

> ### Comment · Reviewer_AzXa · 2024-07-17
> **Following Up**
>
> Hello, Sorry for the delay.
>
> I am generally satisfied with your answers (as well as your response to the other reviewer). The exception to this is Q4/Q5 which I feel that you answered in a very superficial way, without much thought. These are, however, not critical issues, and they will not impact my decision, though I encourage you to think more about these connections.

---

> > ### Author Response · Authors · 2024-07-17
> > **Thank you for your feedback!**
> >
> > Thank you for your feedback! We are delighted to hear that you're generally satisfied with our answers.
> >
> > If you are satified with our response, could you please consider updating the "Claims and Evidence" score?
> >
> > We will also thoroughly address the Q4/Q5 in the final version as you suggested.

---

> > ### Author Response · Authors · 2024-08-07
> > **Follow-Up Again on Your Feedback**
> >
> > Thank you again for your feedback!  We are delighted to hear that you're generally satisfied with our answers.
> >
> > If you are satified with our response, could you please consider updating the "Claims and Evidence" score?
> >
> > We will also thoroughly address the Q4/Q5 in the final version as you suggested.

---

### Review · Reviewer_Xhvb · 2024-05-21

**Summary Of Contributions:**

This paper introduces several novel contributions to the field of privacy-preserving generative models, specifically diffusion models (DMs) that integrate Probably Approximately Correct (PAC) privacy principles. The model incorporates a conditional private attribute guidance module during the Langevin sampling process. This innovative feature enhances the model's ability to target and privatize specific image attributes more effectively, improving the granularity of privacy protection. A new metric for assessing privacy protection in models is introduced. This metric operates by analyzing the output class labels of two nearest neighbor images in the feature space, using a pre-trained classifier, thus providing a more nuanced measure of privacy.

**Audience:**

Yes

**Broader Impact Concerns:**

No additional concerns.

**Claims And Evidence:**

Yes

**Requested Changes:**

Given the weaknesses identified in the paper, I recommend the authors make targeted revisions to strengthen their submission.

**Strengths And Weaknesses:**

Strengths:
1. The primary novelty of this work is the integration of Probably Approximately Correct (PAC) privacy principles into diffusion models. PAC privacy offers a more flexible and potentially more robust framework compared to traditional Differential Privacy (DP). It focuses on the practical difficulty of reconstructing original data from processed data, which is a significant departure from the worst-case scenario focus of DP.
2. This approach allows for a potentially lower noise addition while still ensuring privacy, which can lead to higher utility or quality in the generated images. The concept of leveraging PAC privacy in diffusion models is novel and addresses the utility-privacy trade-off more effectively.

Weaknesses:
1. Overall, the methodology presented in the paper is comprehensive. However, it would be beneficial if the authors could include some theoretical bounds related to Differential Privacy (DP). Providing these theoretical bounds would help in rigorously defining the effectiveness and limitations of the proposed model under the DP framework, thereby enhancing the academic rigor and practical relevance of the research.
2. In the experimental section, the authors have illustrated the potential capabilities of their approach in privacy protection through various charts and graphs. However, the current data presented does not sufficiently support the conclusions drawn in the paper. To strengthen the credibility of their findings, it would be beneficial for the authors to expand their experiments, possibly by including more diverse datasets, comprehensive benchmarks, or additional comparative analyses with existing methods. This enhancement would provide a more robust validation of their claims and better substantiate the conclusions of their study.
3. The overall presentation of this paper still leaves much to be desired, as many sections are not easily understandable. The text contains numerous minor issues, including grammatical errors and problems with formulas. The authors need more time to address these issues to enhance the clarity and accuracy of the document.

---

> ### Author Response · Authors · 2024-05-29
> **Thank you for your comments!**
>
> Thank you for your valuable reviews. We are glad that you think our method is novel and our experiments solid. Below we address your questions one by one. Besides, we have add another baseline, DP-MEPF [1], to our empirical analysis. All revisions are marked blue in the paper.
>
> **Q1. It would be beneficial if the authors could include some theoretical bounds related to Differential Privacy (DP).**
> * We apologize for any confusion. In this paper, we utilized the PAC framework, which differs from the DP framework. Therefore, it is unsuitable to include theoretical bounds related to DP.
> * Nevertheless, the PAC framework has its own bounds controlled by mutual information, as we mentioned in section 3.1, where $\mathcal{M}$ satisfies $(\Delta_{KL}\delta,\rho,D)$ PAC Advantage Privacy if $\Delta_{KL}\delta = D_{KL}(1_\delta||1_{\delta_o^{\rho}}) \leq MI(X;\mathcal{M}(X))$. Moreover, our further experiment in section 5.4 utilizes the bound to determine the noise.
>
>
>
> **Q2. Extra analysis or experiments.**
> * According to your suggestion, we have added another baseline, DP-MEPF [1], to our empirical analysis and found that our model still outperforms DP-MEPF in preserving privacy. Our new results of DP-MEPF are shown in tables below as well as the new Table 1 in the revised paper.
>
> FID for different methods:
> | $\varepsilon$ or $\xi$ | P3DM-Gender | P3DM-smile | DPDM | DPGEN | DP-MEPF |
> | -------- | -------- | -------- | -------- | -------- | -------- |
> | 1        | 175$\pm$2.5| **170$\pm$1.85**  |      |      | **67.5$\pm$1.45**|
> | 5        | 125.8$\pm$3.6| 107.85$\pm$4.7     | 155$\pm$1.5 | 170$\pm$2     | **61.2$\pm$1.8** |
> | 10       | 44.82$\pm$4.48| **37.96$\pm$1.85**| 39.16$\pm$0.68| 117$\pm$1.5| 57.5$\pm$1.2|
> | 15       | 40$\pm$0.5| **37.9$\pm$0.6**| 38.5$\pm$0.73| 113$\pm$1.2|55.8$\pm$1.4|
> | $\infty$ | 34.64$\pm$2.18| 36.03$\pm$1.32| **34.4$\pm$1.8** | 110$\pm$1.2|  52$\pm$2.3 |
>
> Privacy score for different methods:
> | $\varepsilon$ or $\xi$ | P3DM-Gender | P3DM-smile | DPDM | DPGEN | DP-MEPF |
> | -------- | -------- | -------- | -------- | -------- | -------- |
> | 1        | 0.6$\pm$0.05 | **0.63$\pm$0.08**  |      |      |0.48$\pm$0.02|
> | 5        | 0.55$\pm$0.082| **0.6$\pm$0.075** | 0.57$\pm$0.06 | 0.45$\pm$0.03| 0.45$\pm$0.03 |
> | 10       | 0.5$\pm$0.05| **0.56$\pm$0.045**| 0.4$\pm$0.03| 0.47$\pm$0.04| 0.4$\pm$0.04|
> | 15       | 0.45$\pm$0.05| **0.5$\pm$0.025**| 0.4$\pm$0.06| 0.35$\pm$0.05|   0.38$\pm$0.02 |
> | $\infty$ | 0.4$\pm$0.04| **0.48$\pm$0.055**| 0.376$\pm$0.03 | 0.3$\pm$0.06| 0.34$\pm$0.025  |
>
> **Q3. Requirement of improving overall presentation of the paper**
> * Thank you for your suggestion. We've thoroughly reviewed the text to address the minor issues you mentioned, including grammatical errors and problems with formulas. Our goal is to enhance both the clarity and accuracy of the document. We are committed to making the necessary revisions to improve the quality of our paper.
>
> [1] Frederik Harder, Milad Jalali, Danica J Sutherland, and Mijung Park. Pre-trained perceptual features improve differentially private image generation. Transactions on Machine Learning Research, 2023.

---

> > ### Comment · Reviewer_Xhvb · 2024-07-15
> > **To authors**
> >
> > Thank you for your answers. I have carefully read your responses to my questions. The supplementary experiments have proved the effectiveness of your method to a certain extent and can be considered as the results of the strengthened paper.

---

> ### Author Response · Authors · 2024-06-07
> **Follow-Up on Response to Review Comments**
>
> Thank you for your encouraging and constructive comments. We wanted to follow up regarding our previous response to your comments. We would appreciate any additional feedback and welcome any additional questions you might have.

---

### Review · Reviewer_xWeX · 2024-06-14

**Summary Of Contributions:**

Differentially private diffusion model is an important research topic. The challenges can be ensuring robust protection in privatizing specific data attributes.

This paper introduces the PAC Privacy Preserving Diffusion Model. They enhance privacy protection by integrating a private classifier guidance into the Langevin Sampling Process. They introduce a novel metric to gauge privacy levels, and show superior performance in privacy protection  of their models over existing leading private generative models.

**Audience:**

Yes

**Broader Impact Concerns:**

This paper concerns privacy, it doesn't have ethical problems.

**Claims And Evidence:**

No

**Requested Changes:**

1. Introduce all notations clearly.
2. Prove the PAC privacy for the algorithm.
3. Compare with baselines in a fair way.

**Strengths And Weaknesses:**

This paper studies the diffusion model for PAC privacy protection. PAC privacy is a new privacy concept, and the diffusion model has become an advanced generative model.

The writing is not very good for this paper, some notations are not explained, and I have some questions as below.

1. The PAC privacy concept is not clearly stated. Many symbols defined in Section 3 are not explained with their meanings.

2. In Section 4 of the methods, the random response mechanism H, $\tilde{x}$ is the noisy data with Gaussian noise, but it also shows that the probability of H is a random response, which are confusing. It seems that this combines the Gaussian mechanism and the random response mechanism, but they are two different mechanisms. If they can be combined, the authors should explain how to set the parameters. The last sentence of Section 4 is also confusing.

3. What is $\alpha_i$ in Algorithm 1? How to get the PAC privacy of Algorithm 1?

4. For the experiment, how to ensure the fairness of the comparison? Different algorithms use different privacy concepts and get different FIDs. Why set DP-MEPF to blue?

Overall, I don’t think this paper correctly prove their algorithm be PAC private, or prove that their algorithms obtains better experimental results “compared” with the baselines.

---

> ### Author Response · Authors · 2024-06-26
> **Thank you for your reviews!**
>
> Thank you for your constructive comments. Below we address your questions one by one. Besides, we have revised various paragraphs, definitions, notations, and polished the claims in the paper according to your comments. All changes are marked in brown in the revision.
>
> **Q1. The PAC privacy concept is not clearly stated. Many symbols defined in Section 3 are not explained with their meanings.**
> * Thank you for your suggestion. We've thoroughly reviewed the unclear statement of PAC privacy in Section 3.2 and unexplained symbols concept you mentioned in Section 3.2 and 3.3. Besides, other revisions are also included in the paper. We are committed to making the necessary revisions to improve the quality of our paper.
>
> **Q2. In Section 4 of the methods, the random response mechanism H, $\tilde{x}$ is the noisy data with Gaussian noise, but it also shows that the probability of H is a random response, which are confusing. It seems that this combines the Gaussian mechanism and the random response mechanism, but they are two different mechanisms. If they can be combined, the authors should explain how to set the parameters. The last sentence of Section 4 is also confusing.**
> * In the paper DPGEN [1], both the random response and Gaussian mechanisms are used for training.
>   * First, DPGEN injects Gaussian noise with $\tilde{x_i} = x_i + z_i$ given a sensitive dataset $\{x_i : i = 1,2,...,m\}_{i=1}^m$, which serves as reconstructing images with noise conditional score networks [2] leveraged in DPGEN.
>   * Second, after obtaining noisy images, DPGEN utilizes the randomized response method to privatize the training data. The parameter settings for these mechanisms have been modified and updated in the revised paper.
>
> * For last sentence of Section 4, here is a more detailed interpretation, which is also incorporated in the revised paper: we can evaluate the privacy extent of the model by comparing the mean norm $( \mathbb{E}||B||_2 )$ under the same mutual information upper bound level $\nu + \beta$. A smaller mean norm $\mathbb{E}||B||_2$ implies a smaller covariance matrix $\Sigma_B$ and lesser noise addition on $M(X)$, which indicates $X$ and $M(X)$ have less mutual information. Therefore, less noise addition signifies a higher privacy protection of the model $M$.
>
> **Q3. What is $\alpha_i$ in Algorithm 1? How to get the PAC privacy of Algorithm 1?**
> * $\alpha_i$ is the stepsize for annealed Langevin Dynamics.
> * Since the randomized response method adds noise $z_i$ in the random response mechanism, it can be proven to be PAC private using Theorem 2 in the paper. Algorithm 1 is a heuristic method that further enhances privacy. We calculate noise $B$ to measure its privacy strength, achieving the best performance as shown in Table 2. Of independent interest to ensure the PAC privacy of Algorithm 1 itself, we simply need to add noise $B$ to the final $x_T$ in Algorithm 1.
>
>
> **Q4. For the experiment, how to ensure the fairness of the comparison? Different algorithms use different privacy concepts and get different FIDs. Why set DP-MEPF to blue?**
> * To ensure fair comparison, we compare the FID-privacy score patterns of different models using Pareto curves, as shown in Figure 2 of our paper.
> * Specifically, for two models with the same FID, the one with a higher privacy score is considered to have better performance in terms of privacy preservation. For example, if the result curve of model A is above that of model B, model A is considered better compared to model B. Thus, our experimental comparison is fair.
> * DP-MEPF is our new baseline, which was added in the last revision following the request from **Reviewer Xhvb**. Therefore, we set all revisions responding to **Reviewer Xhvb** in blue, including DP-MEPF.
>
>
>
>
> [1] Chen, Jia-Wei, et al. "Dpgen: Differentially private generative energy-guided network for natural image synthesis." Proceedings of the IEEE/CVF Conference on Computer Vision and Pattern Recognition. 2022.
>
> [2] Song, Yang, and Stefano Ermon. "Generative modeling by estimating gradients of the data distribution." Advances in neural information processing systems 32 (2019).

---

> > ### Comment · Reviewer_xWeX · 2024-07-09
> > **Thanks for your response**
> >
> > Afer reading your response, I still have questions about the privacy proof of your algorithm and the fair comparision among baseline.
> > 1. There is no complete proof for the privacy.
> > 2. For experimental comparison,  $\epsilon$ and $\xi$ are mixed used in the Table 1, I'm still confuse here.  You say "for two models with the same FID, the one with a higher privacy score is considered to have better performance in terms of privacy preservation", but in the table, the FIDs are all different.

---

> ### Author Response · Authors · 2024-07-14
> **Thank you for your comments!**
>
> **Q1: There is no complete proof for the privacy.**
>
> Thank you again for your advice on the proof. We've thoroughly reviewed again and completed the proof of PAC privacy in Section 3.2 (with the **complete proof in Appendix A**) and Section 4.4 (with the **complete proof in Appendix B**). Below we summarize some key points:
> * In summary, $D_{f} \big(1_\delta||1_{\delta_o^{\rho}}\big)$ quantifies the divergence between optimal a priori and posterior reconstruction, effectively measuring the difficulty of inference. A higher value of $D_{f} \big(1_\delta||1_{\delta_o^{\rho}}\big)$ signifies greater privacy leakage. Moreover, since $MI\big(X;\mathcal{M}(X)\big)$ provides an upper bound for $D_{f} \big(1_\delta||1_{\delta_o^{\rho}}\big)$, a lower value of $MI\big(X;\mathcal{M}(X)\big)$ indicates stronger privacy protection.}
> * Thus, Theorem 3.1 establishes a general method for linking the difficulty of arbitrary inference (i.e., strength of privacy protection) to the well-known concept of mutual information. With Theorem 3.1, the goal of PAC privacy is explicit: determining (and then lower) the bound $MI\big(X;\mathcal{M}(X)\big)$ with high confidence.
> * Since the randomized response method adds Gaussian noise $Z$ in the random response mechanism, it can be proven to be PAC private using Theorem 3.1 and Theorem 4.1 in the paper. Below we summarize a few key steps (see the full proof in the revised paper):
>     * Algorithm 2 (*which we added during this revision round*) can be combined and written as $H(X) + {B}$, where $H(X)$ represents the RR mechanism and ${B}$ denotes Gaussian noise with ${B} \sim \mathcal{N}(0, \Sigma_B)$.
>     * Applying Theorem 4.1, we can obtain a satisfied upper bound for the whole process $MI(X; \mathcal{M}(X)) \leq \frac{1}{2} \cdot \log \text{det}\big(I_d+ \Sigma_{\mathcal{M}(X)}\cdot \Sigma^{-1}_{{B}}\big)$, where $\mathcal{M}(X) = H(X) + {B}$.
>     * Hence, $MI(X; \mathcal{M}(X))$ can be used to produce a (loose) upper bound of PAC Privacy via Theorem 3.1.
>
> * What's more, Algorithm 1 is a heuristic method that further enhances privacy. We calculate noise $B$ to measure its privacy strength, achieving the best performance as shown in Table 2. Of independent interest to ensure the PAC privacy of Algorithm 1 itself, we simply need to add noise $B$ to the final $x_T$ in Algorithm 1.
>
> **Q2: For experimental comparison, $\epsilon$ and $\xi$ are mixed used in the Table 1, I'm still confuse here. You say "for two models with the same FID, the one with a higher privacy score is considered to have better performance in terms of privacy preservation", but in the table, the FIDs are all different.**
>
> **Difference between $\epsilon$ and $\xi$.** As suggested by the reviewer **AzXa**, $\epsilon$ is for DP methods, and since our P3DM is **not** DP-based, we should not use $\epsilon$ for anything other than a DP parameter. Therefore, we have replaced $\epsilon$ with $\xi$ for non-DP parameters. Specifically, we use $\xi$ exclusively for our P3DM model in the paper.
>
> **Different FIDs.** We are sorry for the confusion. Below we provide several clarifications.
> * Since there is no direct comparison between PAC privacy and DP models, we can only compare them in terms of the trade-off of FID and Privacy Score. This is why we clarified that "for two models with the same FID, the one with a higher privacy score is considered to have better performance in terms of privacy preservation".
> * However, note that in practice, it is nearly impossible to ensure our model has exactly the same FID as other models, because FIDs are compuated as a metric, not a hyperparameter that one can control. As a result, we have to resort to a Pareto plot to demonstrate the trade-off graph of FID and Privacy Score, as shown in Figure 3.
> * Specifically, Figure 3 shows that: For all models with similar FIDs, our P3DM enjoys a higher privacy score and is therefore considered to have better performance in terms of privacy preservation.
> * Note that in Figure 3, the curve that reaches the top-left corner is preferred.

---

> > ### Author Response · Authors · 2024-07-19
> > **Review Comments Follow-Up**
> >
> > Thanks for your encouraging and enlightening comments!  We wanted to follow up regarding our previous response to your comments. We would appreciate any additional feedback and welcome any additional questions you might have.
> >
> > Thank you very much for your time and consideration.

---

> > ### Author Response · Authors · 2024-08-07
> > **Review Comments Follow-Up Again**
> >
> > Thanks for your encouraging and enlightening comments again! We wanted to follow up again regarding our previous response to your comments. We would appreciate any additional feedback and welcome any additional questions you might have.
> >
> > Thank you very much again for your time and consideration.

---

### Decision · Action_Editor_gjdn · 2024-08-14

**Recommendation:** Reject

**Comment:**

I find this to be a borderline case and I recommend the authors to still resubmit a revised version at a later time. Two out of three reviewers are positive about accepting the paper, however the reviewer xWdX and myself are still leaning towards a reject though the paper has improved a lot already during the rebuttal process.

For this decision, there are several reasons, most of them already raised by the reviewers.

Several confusing points have already been corrected during the rebuttal process (e.g., related to the reported $\varepsilon$ values), however I think confusions still remain.

* Similarly to the reviewers, I find Table 1 unclear: why are there both $\varepsilon$ values and $\xi$ values reported? Is $\xi$ the "\varepsilon"-parameter used for the $k$-RR in your algorithm (that is not obvious to me)? For the method by Dockhorn et al., does the reported number refer to $\varepsilon$ or $\xi$? You mention on p. 6 that the method by Dockhorn et al. is not DP as reported, so why would one report the $\varepsilon$-values? I would recommend somehow completely modifying Table 1 or removing altogether.

* When introducing the notion of PAC privacy in Section 3, you write "Furthermore, PAC Privacy also guarantees simple composition bounds, and the automatic analysis framework can be implemented online to analyze composite PAC Privacy loss, even with correlated randomness. In terms of utility, different from DP, the necessary perturbation for PAC Privacy is not lower bounded by $Od)$ for a $d$-dimensional release; instead the bound can be as low as $O(1)$ for many practical tasks, contrasting with the input-independent worst-case information-theoretic lower bound." However, none of these properties are shown. I think it would be important to actually show these properties and also elaborate a bit more on the privacy notion. I think that would be appropriate when introducing a new formal notion of privacy.

* Please be more explicit that the comparisons of Table 2 are empirical. The fact that you use the empirical estimate via your Algorithm 4 to estimate the required level of noise $B$, does not mean that those reported numbers will guarantee the reported bound. Also, why not to write e.g. "estimated level of noise $B$ to approximately ensure PAC privacy", why do you write that bound there? The bound implies the PAC privacy protection, right?

**Audience:**

Definitely, the paper fits very well to TMLR (privacy-preserving diffusion models).

**Claims And Evidence:**

The paper consider training diffusion models for image generation while having the privacy preservation of the training data in mind. The method proposes a novel Langevin diffusion based method for such training and proposes a new notion of privacy, the PAC privacy, the measure how privacy-preserving the final model is. The method is empirically compared to several DP diffusion model training algorithms found from the literature using the FID score and using a novel privacy score, and the results show that the proposed method is superior compared to the baselines. There is a an empirical analysis to measure how PAC privacy preserving the various methods in a sense that how much additional noise is required to obtain a certain level of PAC privacy.

**Resubmission Of Major Revision:**

The authors may consider submitting a major revision at a later time.